# Thunderstorm Activity and Extremes in Vietnam for the Period 2015–2019

**Khiem Van Mai [1], Terhi K. Laurila [2],*, Lam Phuc Hoang [1], Tien Duc Du [1], Antti Mäkelä [2] and Sami Kiesiläinen [2]**

[1] Viet Nam National Center for Hydro-Meteorological Forecasting, Viet Nam Meteorological and Hydrological Administration, 8 Phao Dai Lang Str., Dong Da, Hanoi 100000, Vietnam
[2] Finnish Meteorological Institute, Erik Palménin Aukio 1, FI-00560 Helsinki, Finland
* Correspondence: terhi.laurila@fmi.fi

**Abstract:** Within a meteorological capacity building project in Vietnam, lightning location data and manual (human-observed) thunderstorm day observations were analyzed for the period 2015–2019. The lightning location dataset, based on the global lightning detection system Vaisala GLD360, consists of a total of 315,522,761 lightning strokes. The results indicate that, on average, 6.9 million lightning flashes per year occur in the land areas of Vietnam; this equals a lightning flash density of 20 flashes km$^{-2}$ yr$^{-1}$. The largest average annual flash density values occur in three regions in North, Central and South Vietnam. The majority of lightning occurs in the monsoon season (April–September), peaking in May, while in October–March, the lightning activity is very modest. During individual intense thunderstorm days, the flash density may exceed 12 flashes km$^{-2}$ day$^{-1}$. Thunderstorms in Central Vietnam are generally more intense, i.e., more lightning is expected on average per one thunderstorm day in Central Vietnam than in other regions. This study is a continuation of several years of meteorological capacity building in Vietnam, and the results suggest that large socio-economic benefits can be received by understanding the local thunderstorm climatology in high detail, especially in a country such as Vietnam, where lightning causes substantial socio-economic losses annually.

**Keywords:** extreme weather; thunderstorms; lightning; thunderstorm day; lightning location; thunderstorm climate

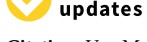



## 1. Introduction

Thunderstorms and their hazards, such as lightning, affect Asian societies in many negative ways. Lightning-related death rates are generally several times greater in Asian countries than those in Europe or North America [1]. Despite the fact that lightning-related fatality rates have globally decreased over the past 100 years or so, especially due to urbanization [1], societies will in practice always be significantly affected by lightning because of its sudden nature. Furthermore, the accurate forecasting of tropical convection and related thunderstorms is one of the greatest challenges in numerical weather prediction (e.g., [2]).

Although lightning is nowadays among the essential climate variables in meteorology [3], its observational homogeneous datasets are somewhat limited globally. Detailed observations with lightning location systems exist at present, but the datasets temporally cover 20 years at best, which is quite a limited sample, especially in climatological studies and trend analyses. The recent technical enhancements in global lightning detection with satellites [4–6] has made the collection of detailed information on lightning possible, but temporally extensive datasets are available only in the future.

In climatological studies, two thunderstorm-related variables have been used: the amount of lightning (per unit area and time) and the number of thunderstorm days. The former needs the counting (but optimally the locating) of individual lightning flashes of a thunderstorm as a minimum, which in the past has made this variable challenging

to acquire from large areas with good accuracy. The latter, however, is simpler because, for one thunderstorm day, only one flash per day needs to be observed [7]. Indeed, a thunderstorm day (or thunder day) is the most historical thunderstorm variable measured globally in many countries, but unfortunately, the systematic synthesis of these data was made a very long time ago [8]. Our study is an attempt to show how the thunderstorm day observations, in our case from Vietnam, can be analyzed as monthly and annual summaries and compared with lightning location system observations.

Scientific studies have been made about lightning activity for nearby areas of Vietnam, e.g., for Bangladesh [9], the Tibetan Plateau [10], Sri Lanka [11] and regions in China [12,13]. Recently, the multi-annual variability of Southeast Asian thunderstorms has also been examined [14]. However, the thunderstorm characteristics of Vietnam have only been analyzed in a few studies. The Institute of Geophysics at the Vietnam Academy of Science and Technology has operated various atmospheric electricity measurement systems in Vietnam, including a lightning observation system [10]. Based on the measurement data, statistics of the lightning occurrence in Vietnam have been reported in non-peer-reviewed Vietnamese publications. Reference [15] also presents lightning statistics from Vietnam for the period 1998–2014 based on satellite data (i.e., the Lightning Imaging Sensor, LIS; [16,17]). Recently, ref. [18] studied the lightning warnings in the Hanoi region based on various datasets, including electric field observations of thunderstorms. Even based on the results published in these few publications, it is evident that a substantial amount of lightning occurs in Vietnam. Therefore, the exact spatio-temporal analysis of lightning is of interest as it supports, for example, the overall safety of people and property and the planning of critical infrastructure.

The Ministry for Foreign Affairs (MFA) of Finland has funded capacity development projects in many regions around the world in recent years. Many of the projects have focused on the development of the hydro-meteorological readiness and capability of the National Hydro-Meteorological Services (NHMS), where the Finnish Meteorological Institute (FMI) has provided the expertise for hydro-meteorology. The projects have contained a wide range of topics, e.g., data collection systems, forecasting, climate services and IT-infrastructure [19,20]. In Vietnam, FMI has co-operated with the Vietnam Meteorological and Hydrological Administration (VNMHA) since 2010. A capacity building project stream in Vietnam entitled "*Promoting Modernization of Hydrometeorological Services in Vietnam*" (PROMOSERV) is now in phase 3. One specific activity has been to train the VNMHA personnel to use and understand the applicability of lightning location observations and their usability in providing a real-time lightning location data feed. Co-operation continues in PROMOSERV3 within which this lightning study is also situated.

Located in the geographical area between latitudes 8° to 23° N, the weather in Vietnam is affected by both extratropical and tropical systems [21]. Highly variable terrain and a long coastline (over 3200 km adjacent to the South China Sea—East Sea of Vietnam or Bien Dong) lead to a large climatological variation, including sub-climate regions from the North to South and from East to West [22–24]. Northern Vietnam (from approximately 20° to 23° N) has a tropical monsoon climate with four distinguishable seasons characterized by high humidity, while the climate of southern Vietnam (from approximately 8° to 12° N) is a moderate tropical climate with clearly separate dry and rainy seasons [22]. Central Vietnam (from 12° to 20° N) is characterized by complex-narrow topography, and its climate is the combination of the northern and southern climates.

The weather in Vietnam is dominated by the East Asian Monsoon. In winter or in the dry season (November to April), the monsoon winds blow from the northeast along the coast of China and across the Gulf of Tonkin, bringing cool and dry air from the northeast. Consequently, the winter season in most parts of the country is dry in comparison with the rainy or summer season [22]. During the rainy season (May to October), approximately 80% of the precipitation is related to the southwesterly summer monsoon, tropical cyclones from Bien Dong, and other tropical disturbances. Annual rainfall ranges from 1000 to 3000 mm, and temperatures range from as low as 5 °C in December–January (i.e., the cold months) to

above 37 °C in June–July (i.e., the hot months). Except for the highlands, seasonal variations are clearer in the northern half of the country, whereas seasonal temperatures vary only by a few degrees (usually between 21 and 28 °C) in the southern half of the country [25].

In this paper, t the characteristics of the lightning occurrence in Vietnam in the period 2015–2019 are presented based on lightning location data (Vaisala GLD360). This study forms the basis for VNMHA to analyze and interpret lightning location data to examine the local thunderstorm activity. Furthermore, besides the analysis of lightning location data, this paper also presents a summary of the thunderstorm day observations based on the human observations at the synoptic weather stations of VNMHA.

This paper is organized as follows. The data and the analysis methods are summarized in Section 2. Section 3 presents the results with examples of the most intense thunderstorm episodes during the study period. Section 4 provides the discussion and conclusions, respectively.

## 2. Data and Methods

### 2.1. Lightning Location Data

Vaisala Inc. operates a global lightning location network GLD360 [26,27]. The working principle of GLD360 is basically similar to any ground-based lightning location system, but the sensors are distributed around the world to achieve global coverage. The network sensors locate the very low frequency (VLF) electromagnetic signal of lightning return strokes and send the signal information to the central processing unit situated in the United States. The central processor then combines the received sensor information in real-time, calculates the most probable lightning strike point, and sends the information of the return stroke (such as time and location and other information such as peak current with polarity) to the end-users. The basic output unit is a stroke, either a cloud-to-ground (CG) or intracloud (IC) stroke. At present, GLD360 classifies the strokes as CG or IC, but in our dataset, the classification is available only in 2019. However, in 2019, the percentage of IC strokes is only 2.7%, which suggests that the detection efficiency (DE) of GLD360 for IC strokes is quite low. This is because, in the latitudes of Vietnam, there should be several times more IC strokes than CG strokes [28]. Although the classification is available only for 2019 data, the other years presumably include a similar percentage of IC strokes. Therefore, all years in our dataset (2015–2019) are considered homogeneous.

According to previous studies (e.g., [29]), the true nature of weak positive strokes (peak current < +10 . . . 15 kA) in medium-range (LF) lightning location systems is questionable. Weak positive strokes are suggested to be in reality IC strokes since IC strokes have signals resembling weak positive lightning. In some studies, these events are removed from the data [30]. However, for a long-range VLF-system, such as GLD360, there are no specific studies on this topic to our knowledge. Therefore, the low-peak current strokes are not removed from the data in this study. However, we acknowledge that our data may contain misclassified IC-strokes to some extent but finding out their actual number would require further analysis.

The data can be organized as flashes according to commonly used standards (i.e., using spatio-temporal criteria for adjacent strokes). In this study, the strokes are organized as flashes using a 1.0 s time threshold and a 15 km distance threshold: this means that all strokes that occur within 1.0 s and 15 km are organized as one flash. In this study, the peak-current statistics are also presented.

The performance of a lightning location system is usually expressed with DE, which is the ratio between the number of the observed and actually-occurred lightning. The operator of GLD360 has stated that the DE for CG flashes is 70% globally, and this has been noted to be a good estimation, at least in Europe [26,31]. The examination of DE has been established also in North America [32,33] but to our knowledge, there are no performance studies in Vietnam or nearby regions. However, the peak current distribution of GLD360 data in Vietnam (see Section 3.4) suggests that the DE for low-peak current strokes (less than about 5 kA) is quite low, as has also been noted by ([26], their Figure 5). This suggests

that the overall DE of GLD360 is clearly below 100%. On the other hand, because the data also contain <10 kA strokes to some extent, the DE must be relatively high. Therefore, the 70% DE for CG flashes given by the GLD360 operator is a valid assumption.

The lightning location data from the period 2015–2019 contains a total of 315,522,761 strokes in the study area 7–25° N, 100–112° E (see Figure 1). These are further organized as flashes with the above-mentioned thresholds. For the spatial analysis, a density map is produced as follows: first, the individual lightning locations (i.e., flashes) are analyzed in a 1 × 1 km grid. Then, the flash density is smoothed using a Gaussian convolution: the Gaussian convolution window is a two-dimensional gaussian function with a standard deviation of 5 in a 50 × 50 km grid. Lastly, the flash density is scaled to a unit of flashes per 100 km$^2$. We note that the selection of the scaling to 100 km$^2$ is arbitrary, but it has been noted to be a good selection for lightning density maps because it is sufficiently small and large at the same time (i.e., a larger scaling area would give a too smoothed map, while a smaller scaling area would give a too noisy field; [34]).

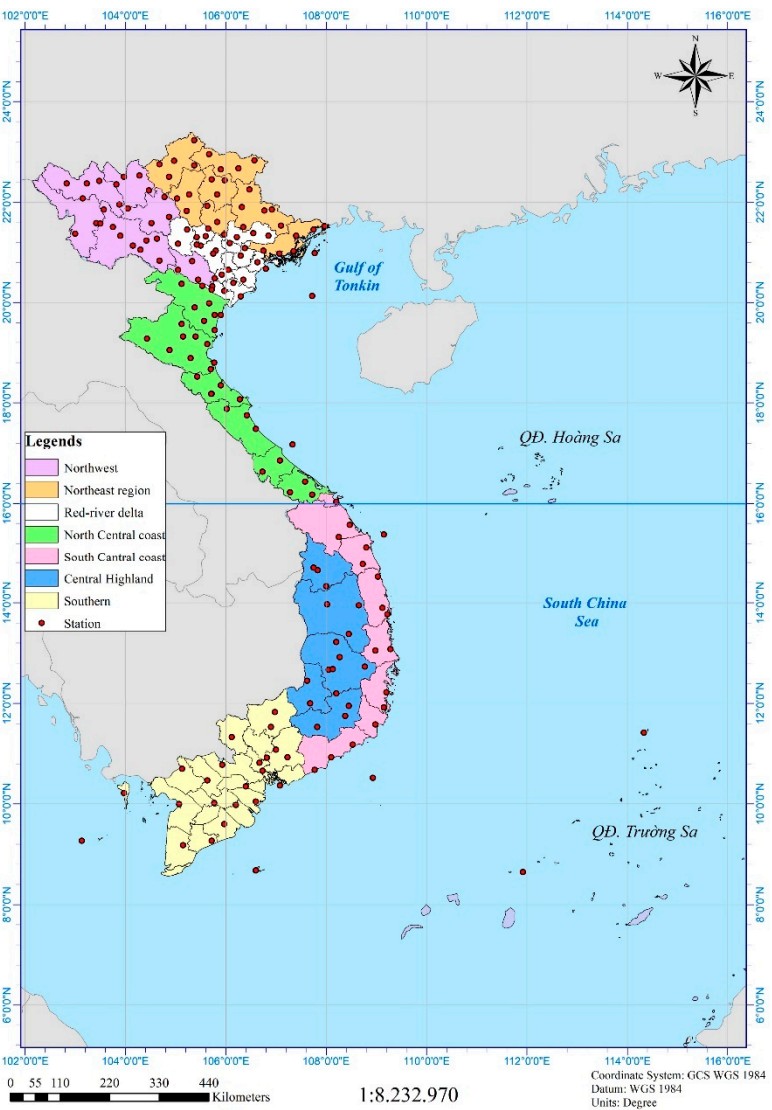

**Figure 1.** The map of the study region, land areas of Vietnam, the Vietnamese synoptic observation stations (brown dots) over different sub-climate regions, and the division of Vietnam into North and South Vietnam (horizontal line at 16° N).

The spatial distribution of thunderstorm days based on GLD360 data was computed to compare it with human-observed thunderstorm days. The thunderstorm days were

calculated for a $20 \times 20$ km grid since it is a valid estimated area (i.e., around 11 km$^2$ radius) for a human to be able to observe thunderstorms.

Because the main interest in this study is the spatio-temporal occurrence of lightning in Vietnam, the data are also masked specifically for the land areas of Vietnam. This dataset is used for analyzing the temporal flash counts (annual, monthly, and daily) in the whole of Vietnam and in North and South Vietnam (separated by the 16° N line. Additionally, the most lightning-abundant days are selected for closer examination.

### 2.2. Thunderstorm Day Data

A thunderstorm day (or thunder day) is the most historical variable for observing and tracking the occurrence of thunderstorms. According to the World Meteorological Organization, the definition of a thunderstorm day is that thunder is observed at a synoptic observation (weather) station during a day [7]. Because 1 and, e.g., 1000 thunder occurrences per day both equal one thunderstorm day, the variable itself does not provide information about the intensity of the storm. Nonetheless, it is a very informative variable for indicating the overall occurrence of thunderstorms, especially because many countries have observational datasets covering several decades or even more than a hundred years.

For this study, the thunderstorm day observations have been analyzed for the period 2015–2019 from a total of 184 synoptic stations (plotted in Figure 1). The station-based observations have been analyzed as annual and monthly averages for the whole of Vietnam and separately for North and South Vietnam (divided by the 16° N line). By dividing Vietnam at the 16° N line, North Vietnam is around 176,800 km$^2$ and South Vietnam 170,400 km$^2$.

## 3. Results

### 3.1. Thunderstorm Occurrence in Vietnam

The annual numbers of lightning flashes in 2015–2019 in Vietnam and separately in North and South Vietnam are shown in Figure 2. The lowest number of lightning flashes in Vietnam occurred in 2015, with around 4.6 million flashes, and the highest number in 2019, with around 9.3 million flashes. In 2016–2018, the flash counts were almost the same (about 7 million flashes per year). It must be noted that five years is too short a time period to investigate any possible trends. The annual mean and median flash number in 2015–2019 was 6.9 million flashes per year. Considering the surface area of Vietnam of about 347,200 km$^2$, the average annual flash density is 20 flashes km$^{-2}$ yr$^{-1}$. When considering North and South Vietnam separately, there are no evident differences that either region would have a substantially greater annual flash count than the other (Figure 2).

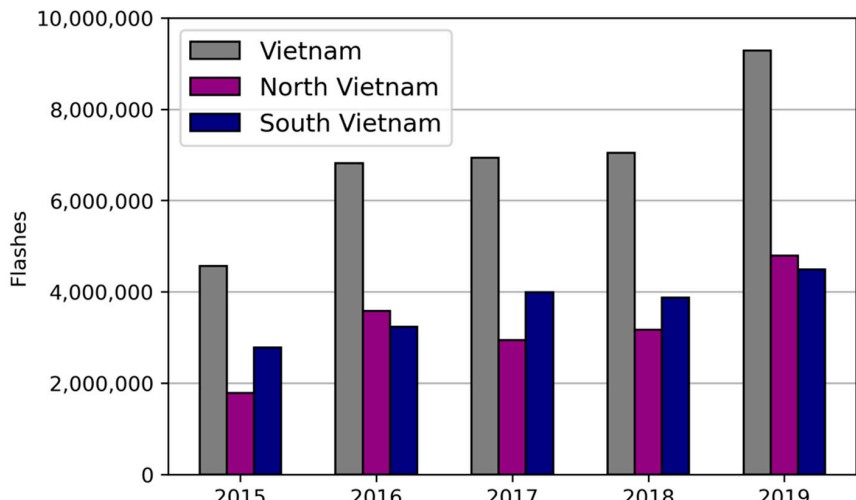

**Figure 2.** Annual number of lightning flashes in the whole of Vietnam, North Vietnam, and South Vietnam in 2015–2019.

The monthly means of lightning flashes in Vietnam show two peaks in lightning activity (Figure 3): the first from April to June and the other from August to September. The first peak is associated with the transition season in the North and the summer (Southwest) monsoon season in the South. In the transition season, the temperature increases and thunderstorms often occur when cold air from mid-latitudes moves to Vietnam. The Southwest monsoon season often starts in early May in South Vietnam, which brings more rainfall and thunderstorms to the area. The thunderstorm season from April to June is clearly more active in lightning than the season in August and September. The largest amount of lightning distinctly occurs in May, with over 1.7 million lightning flashes, which is almost 25% of the annual mean. The least amount of lightning occurs from December to February. The two thunderstorm seasons are also seen in North and South Vietnam. The lightning in May, September, October and November occurs more in the South, whereas in August in the North. Other months have a relatively even number of flashes between North and South. Especially in the autumn, lightning occurs more in South Vietnam.

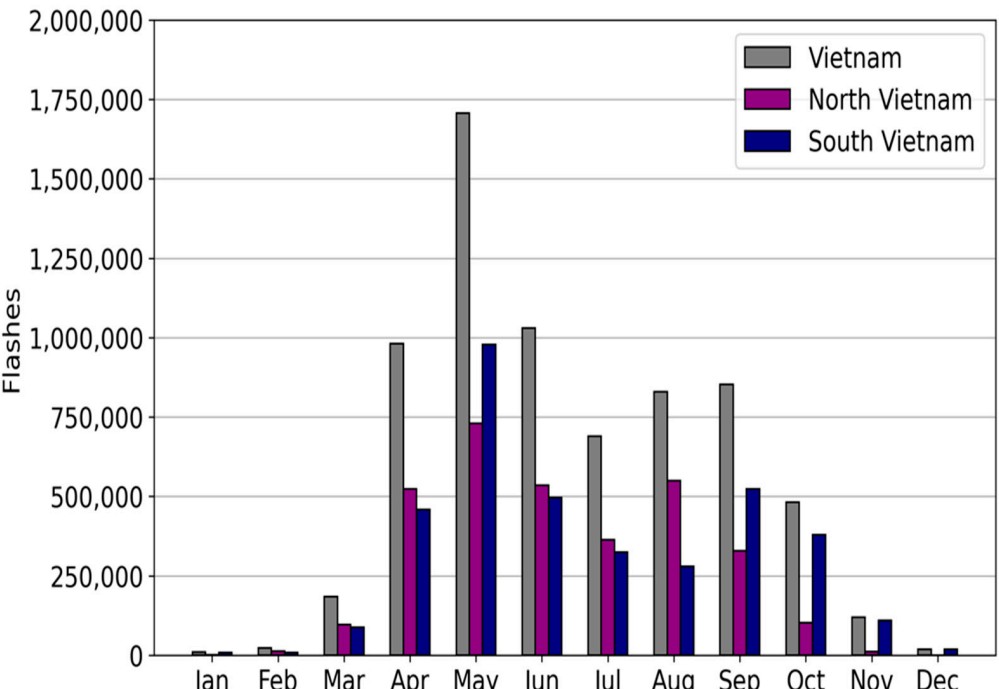

**Figure 3.** Monthly means of lightning flashes in the whole of Vietnam, North Vietnam, and South Vietnam in 2015–2019.

The daily mean flash numbers (Figure 4) reveal the same thunderstorm season peaks in late spring—early summer and autumn that are visible in the monthly mean flashes (Figure 3). The highest daily flash number in Vietnam is, on average, around 100,000 flashes, and the maximum daily flash number is over 250,000 flashes (Figure 4a). In the North, the two main thunderstorm seasons are relatively similar in terms of the number of lightning flashes (Figure 4b). In contrast, in the South, the earlier season has a higher lightning flash number than the autumn season. The autumn thunderstorm season in the North is shorter and occurs earlier than in the South.

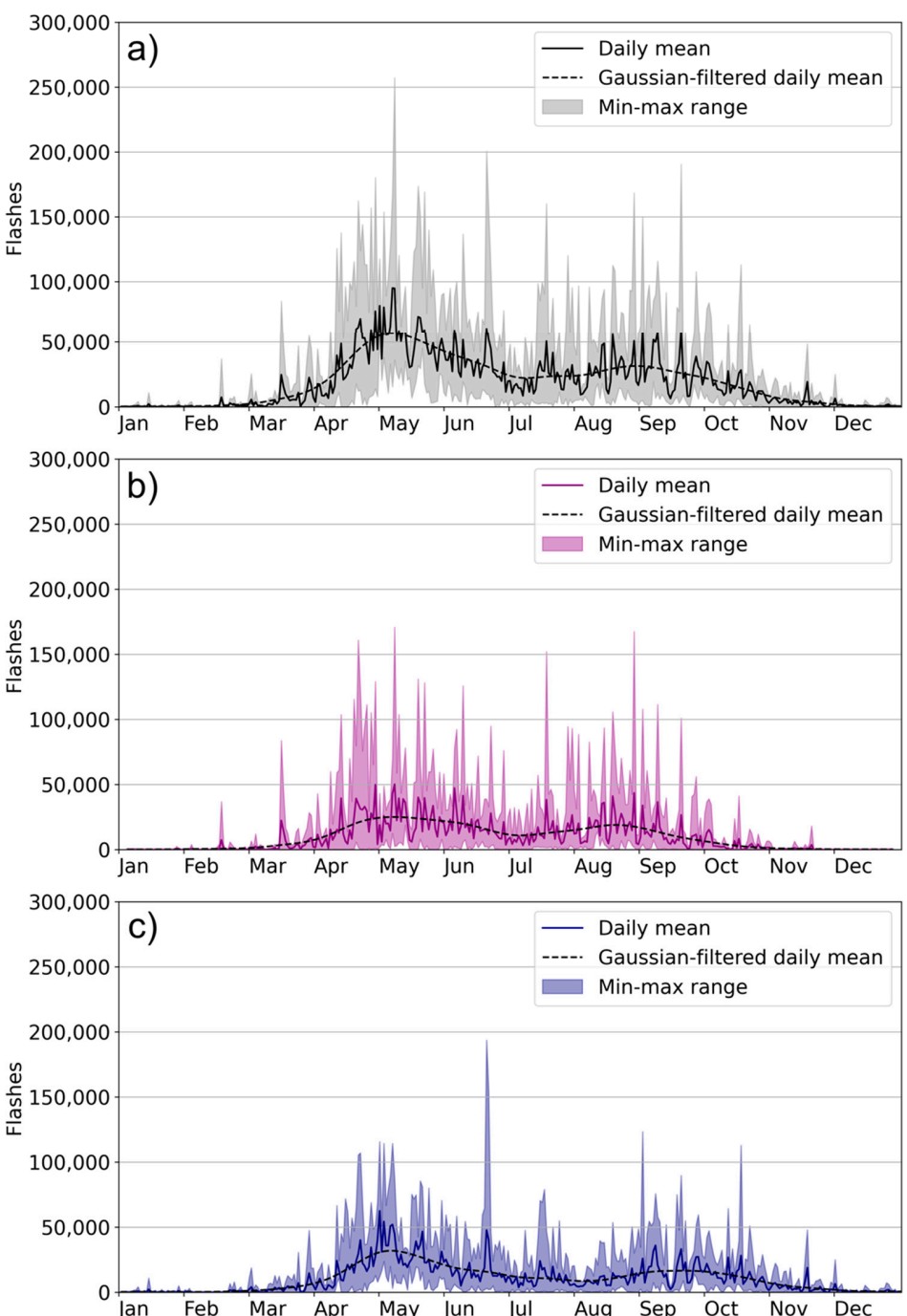

**Figure 4.** Daily means (solid lines), Gaussian-filtered daily means (dashed lines) and min–max range (shaded colors) of lightning flashes in (**a**) the whole of Vietnam, (**b**) North Vietnam, and (**c**) South Vietnam in 2015–2019.

The annual mean lightning density shows three regions where most of the lightning occurs in Vietnam (Figure 5): the first in the northeast part of Vietnam around and north to 20° N, the second in the central part of Vietnam between 14 and 17° N, and the third at the most southern part of Vietnam south to 12° N. The annual lightning density maps for each individual study year (not shown) indicate the same regions, which suggests that the regions of high lightning density observed in Figure 5 are more or less common every year.

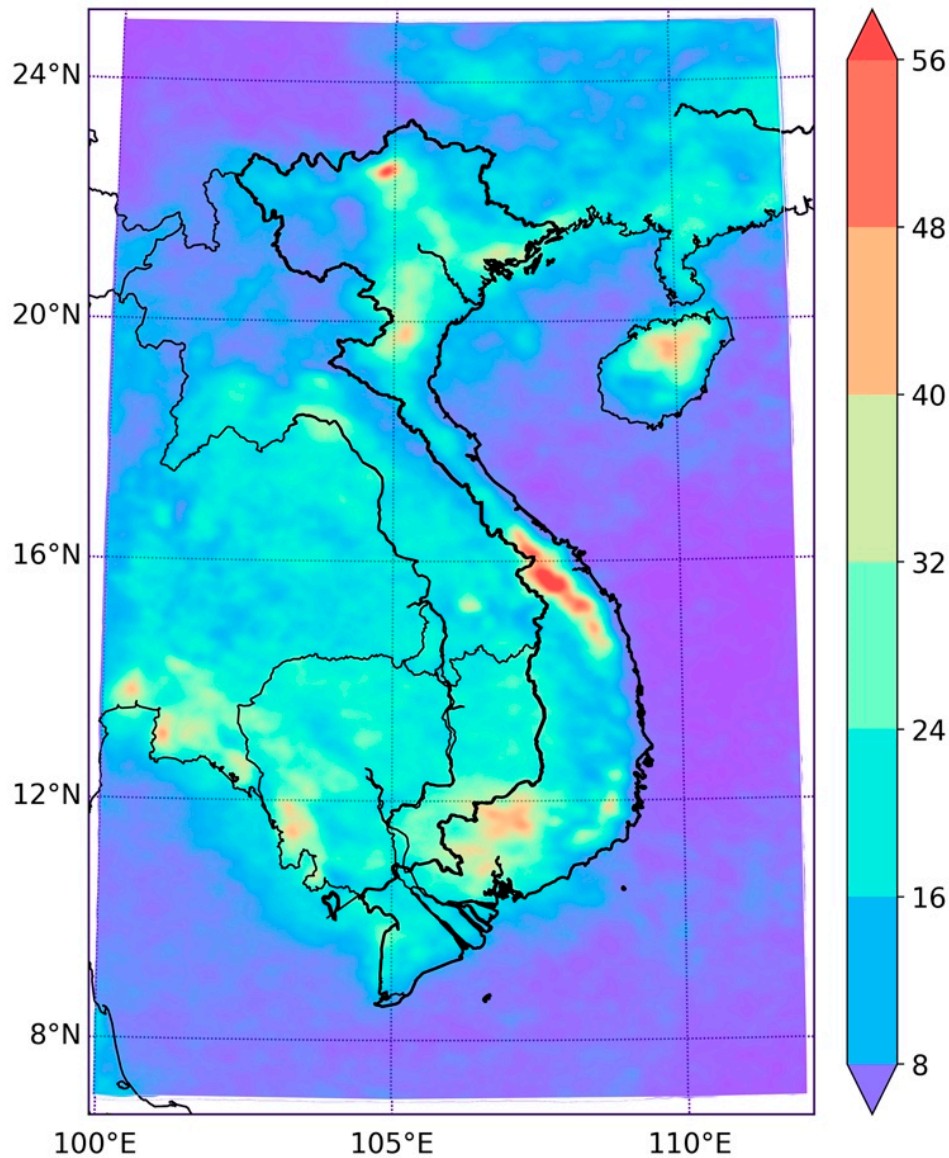

**Figure 5.** Annual mean lightning density (flashes $km^{-2}$ $yr^{-1}$) for the period 2015–2019.

*3.2. Thunderstorm Days*

The annual and monthly mean number of thunderstorm days observed by the synoptic weather station network of Vietnam during the period 2015–2019 are shown in Figure 6. The values indicate the mean number of days per year when thunder is locally observed. The lowest number of thunderstorm days in Vietnam, 65, was observed in 2015, whereas in 2017 the number was 74. During these five years, the annual number of thunderstorm days increased from 2015 to 2017, then decreased in 2018 and 2019. When considering the number of thunderstorm days in North and South Vietnam separated by 16° N, the number of thunderstorms occurring in North Vietnam was the same as those occurring in South Vietnam in 2018 and 2019. During 2015–2017, the number of thunderstorms occurring in South Vietnam was always higher than that of thunderstorms occurring in North Vietnam.

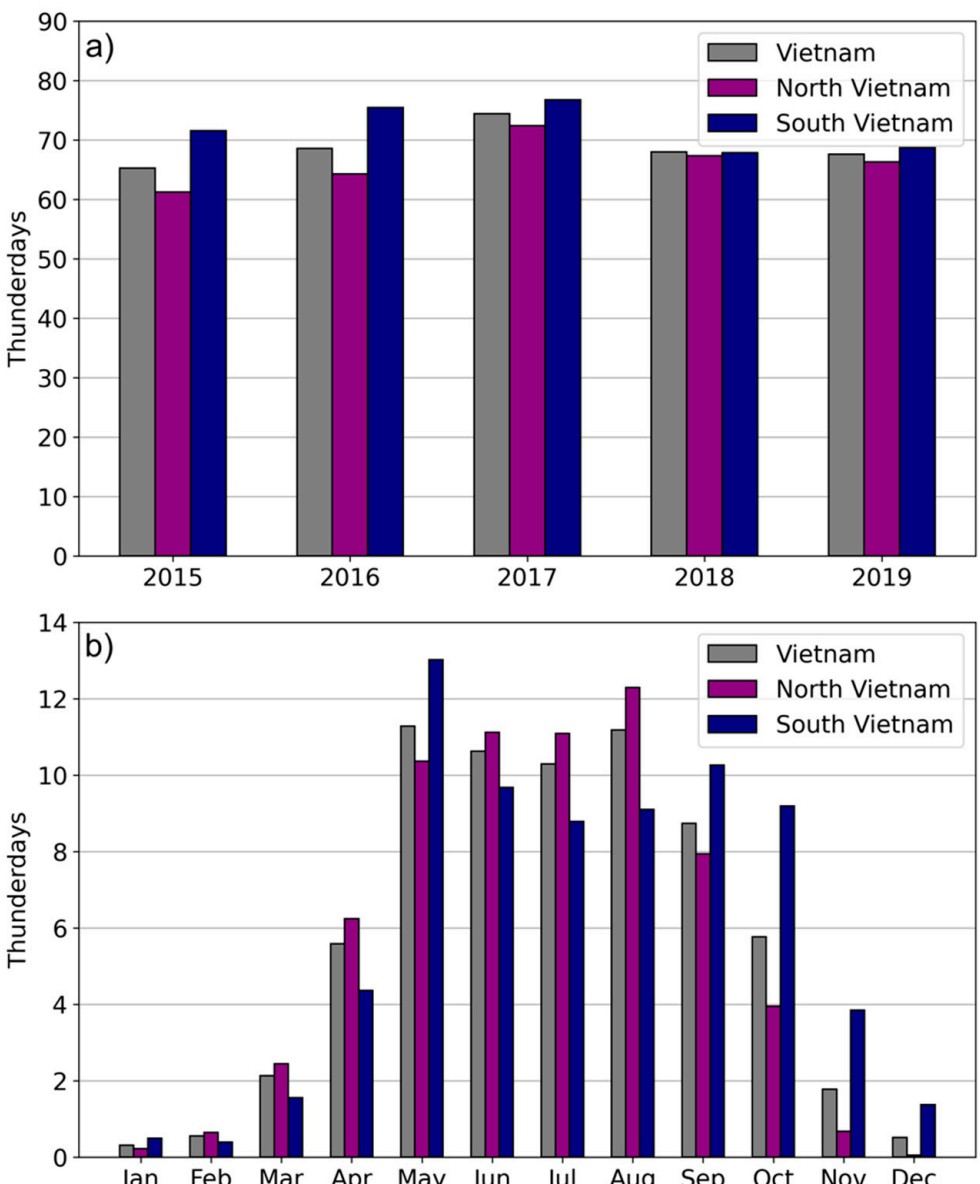

**Figure 6.** The annual (**a**) and monthly (**b**) mean number of thunderstorm days in North, South and the whole of Vietnam from 2015 to 2019 according to manual observations plotted in Figure 1.

In the monthly mean distribution (Figure 6b), the largest number of thunderstorm days in Vietnam is observed in May and August, although the numbers are high also in June and July, which is consistent with the rainy season in Vietnam. The number of thunderstorm days in the summer and autumn seasons is higher than in the spring and winter seasons. From September to December, the number of thunderstorm days is much larger in South than in North Vietnam: this pattern is also seen in the monthly distribution of lightning (Figure 3).

The mean spatial distribution of observed thunderstorm days in 2015–2019 is shown in Figure 7. While the distribution of observed thunderstorm days highlights the same three regions that have high lightning densities (Figure 5), the intensities of these regions are somewhat different. The highest lightning density region is in Central Vietnam, whereas the highest number of thunderstorm days occurred in South Vietnam. This suggests differences in the intensity of the thunderstorms in these regions, i.e., the high lightning densities in Central Vietnam occur with fewer thunderstorms. In contrast, in South Vietnam, the thunderstorms are, on average, associated with less lightning.

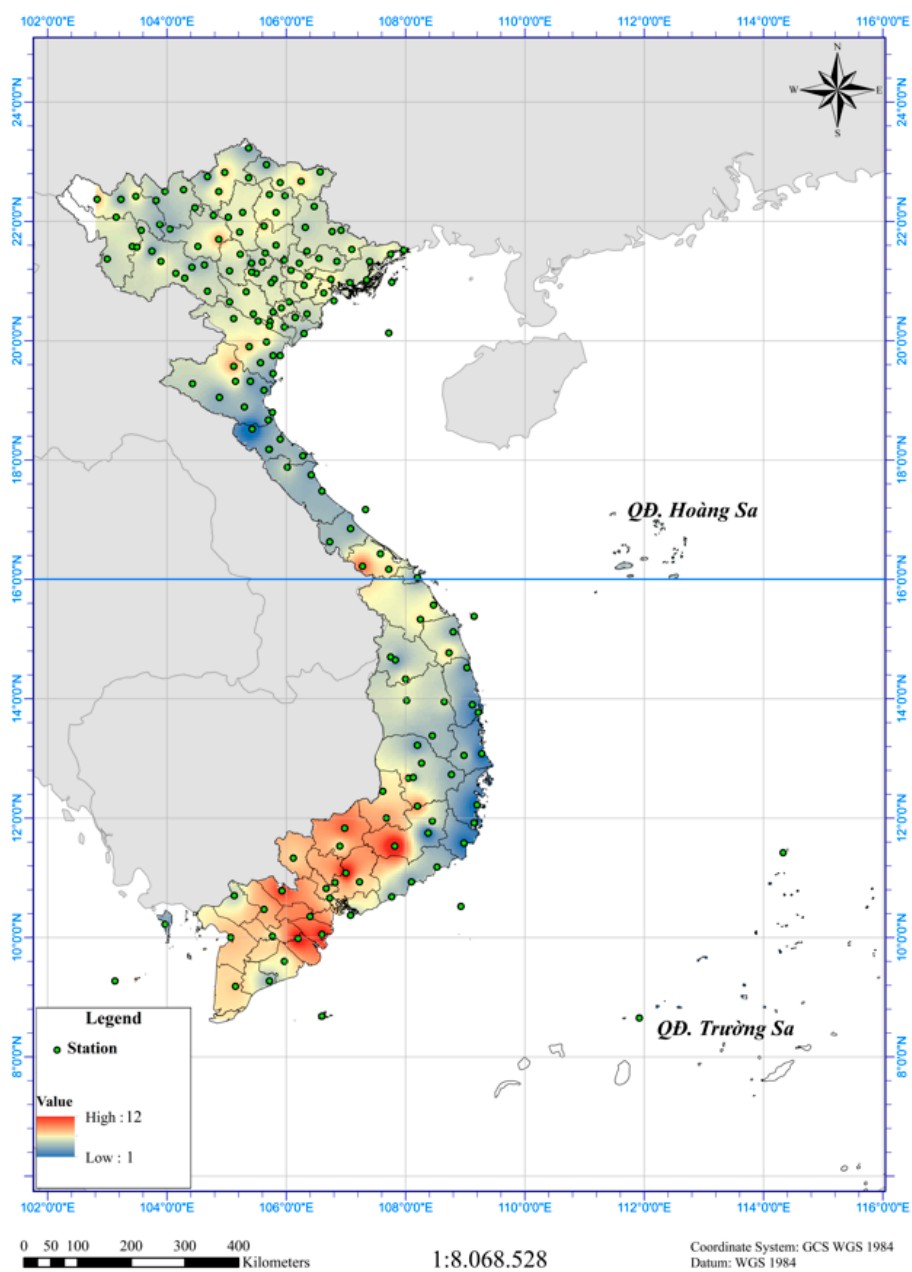

**Figure 7.** The average annual number of thunderstorm days in Vietnam in 2015–2019 based on human observations.

The thunderstorm days calculated based on the GLD360 data are presented in Figure 8. When compared to the human-observed thunderstorm days (Figure 7), it can be concluded that the spatial distributions are largely similar. The highest number of thunderstorm days, which in both data sources occurred in South Vietnam, is around 180 based on GLD360 and around 140 based on human observations. It is obvious that the number and distribution of observation stations has an effect on the thunderstorm day results based on observations. As noted in Section 2.1, the human-observable radius of thunder is around 10 km. Hence, it is likely that not all thunderstorms are observed since the observation stations do not cover the whole country in a 10 km grid (see locations of the stations in Figure 7). Therefore, a lower number of thunderstorm days in the observations compared to the GLD360 data is expected.

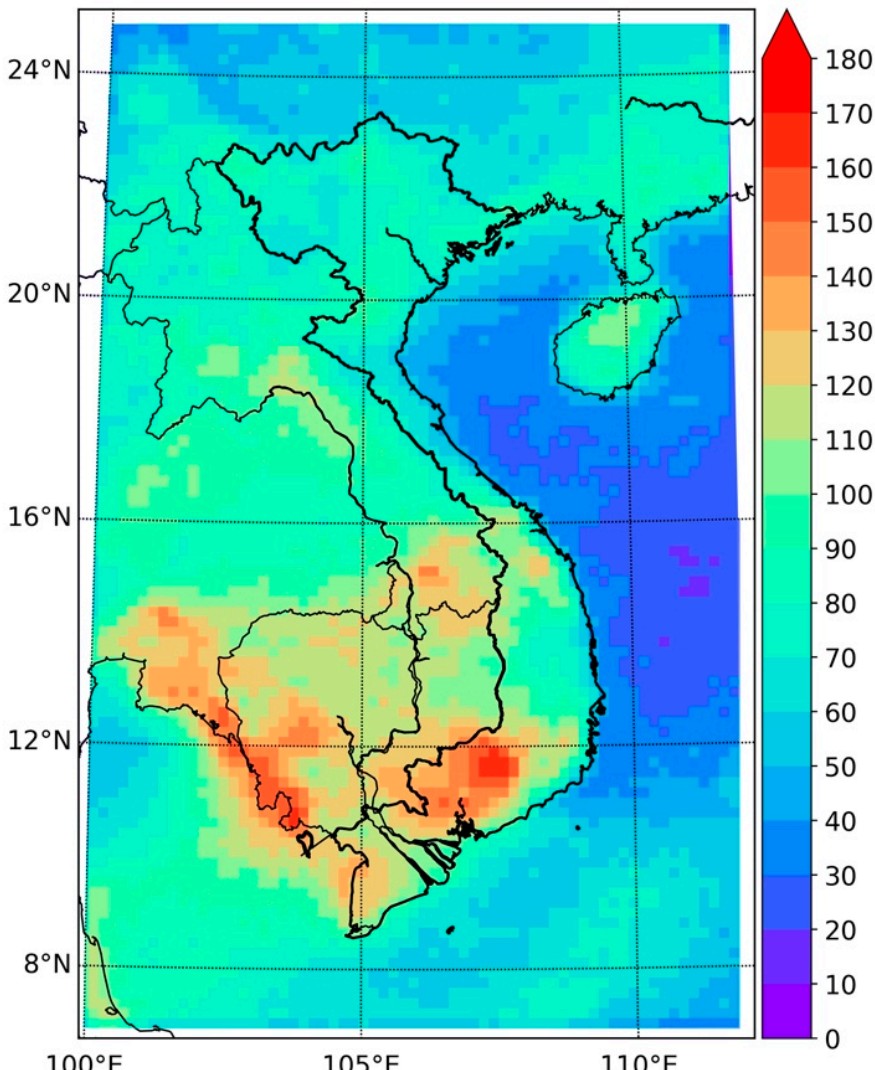

**Figure 8.** The average annual number of thunderstorm days in Vietnam in 2015–2019 based on GLD360 data.

### 3.3. Most Intense Thunderstorm Days

Table 1 shows the ten most intense thunderstorm days in Vietnam during the study period. Here, the intensity is represented by the number of daily located lightning flashes. The most lightning occurred on 9 May 2018 with almost a total of 260,000 flashes. In Figure 9, the daily flash density maps are shown for the three most lightning-abundant days in Table 1. These three cases differ in the spatial occurrence of lightning. Thunderstorms on 9 May 2018 (Figure 9a) are distributed practically all around Vietnam land areas. There is a clear land-sea contrast suggesting the occurrence of typical air mass thunderstorms. The thunderstorms were formed ahead of a front in northern Vietnam. In the case of 21 June 2019 (Figure 9b), the thunderstorms occurred in central and southern Vietnam and in the case of 20 September 2015 (Figure 9c), the thunderstorms occurred in northern and southern Vietnam. Lightning occurred in both of these cases in widespread convective systems producing locally extremely large amounts of lightning and both cases are associated with cyclonic circulations in northern (2015 case) and southern (2019 case) Vietnam.

**Table 1.** The ten most lightning-abundant days in Vietnam in 2015–2019.

| Date | Flashes |
|---|---|
| 9 May 2018 | 257,184 |
| 21 June 2019 | 200,525 |
| 20 September 2015 | 190,516 |
| 29 April 2016 | 180,168 |
| 20 May 2019 | 173,561 |
| 23 May 2017 | 169,154 |
| 8 May 2018 | 168,715 |
| 29 August 2019 | 168,346 |
| 21 April 2016 | 162,207 |
| 19 July 2019 | 160,073 |

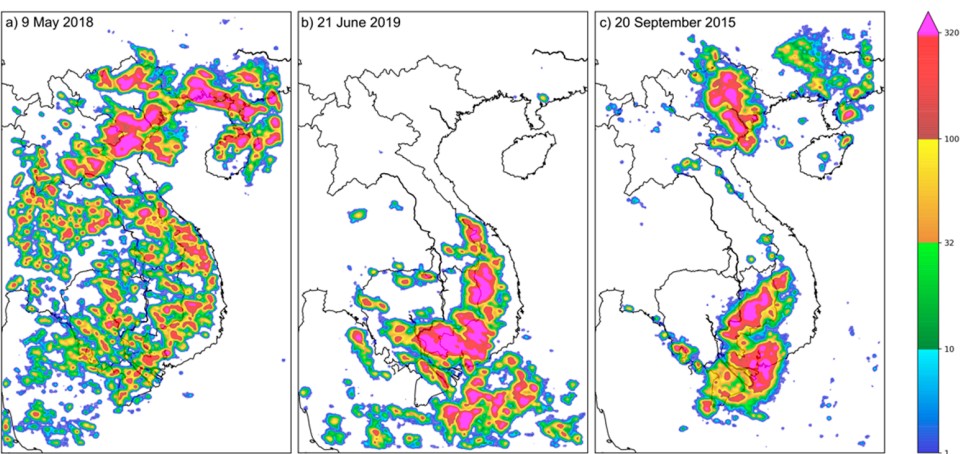

**Figure 9.** Lightning density maps for the three most lightning-abundant days in the study period: (**a**) 9 May 2018 (257,184 flashes), (**b**) 21 June 2019 (200,525 flashes), and (**c**) 20 September 2015 (190,516 flashes). The values are flashes $100 \, \text{km}^{-2} \, \text{day}^{-1}$.

*3.4. Peak Currents*

Figure 10 shows the peak current distributions for negative and positive flashes. The results show that the majority of peak currents are below 20 kA. The mean peak currents are 12 kA for positive flashes and −16 kA for negative flashes.

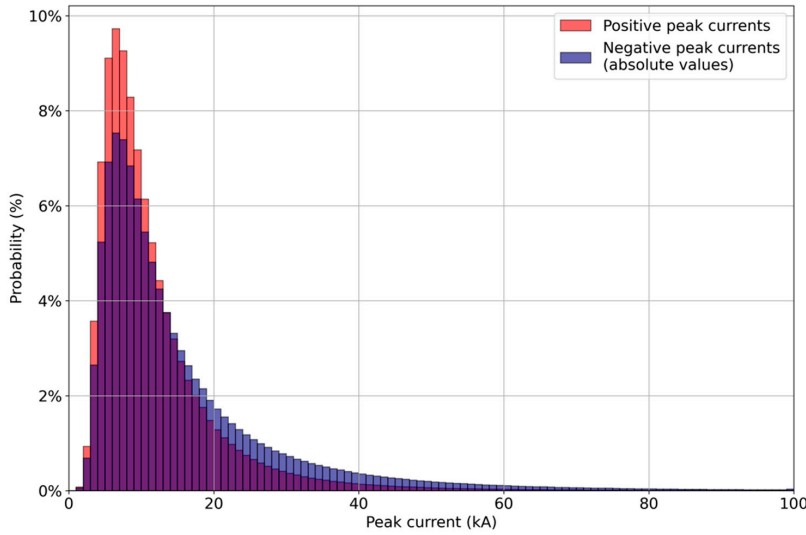

**Figure 10.** Distributions of peak current for positive (red) and negative (blue, as absolute values) flashes in Vietnam in 2015–2019. The x-axis is limited to 100 kA.

As discussed already in Section 2, the peak current distribution can be used for examining the lightning location system performance. This is because the probability of observing the lightning electromagnetic signal is governed by the signal amplitude, which in turn depends on the return stroke peak current. Because the signal amplitude dissipates with distance, the lower the peak current, the more unlikely it is that the signal will be observed. Of course, the sensor geometry (i.e., the number of sensors in the network and their relative distances) and the random occurrence of lightning within that geometry plays a role. To conclude, for any lightning location system, typically the DE is larger for higher peak current strokes, while for lower peak currents (say, below 5 kA), the DE may drop substantially.

Based on the GLD360 peak current distribution and comparing that with distributions of other networks in previous studies, we can hypothesize that small peak currents are not fully represented in the distribution of GLD360. Therefore, the DE for these peak currents is less than for larger peak currents.

## 4. Discussion and Conclusions

Weather hazards have been causing increasing impacts on societies during the past decades all around the world (e.g., [35]). In Vietnam, extreme weather events are also estimated to be more frequent and intense [22,24]. Among the various hazardous weather events in Vietnam, lightning is one of the most dangerous: according to [1], the annual lightning death rate is 1.2 to 8.8 per million people depending on the region. This is somewhat not surprising considering the results of our study. The average annual lightning flash density values locally of up to 63 flashes $km^{-2} yr^{-1}$ and daily values greater than $12 km^{-2} day^{-1}$ in some areas indeed suggest substantial impacts of lightning on people, cattle, and infrastructure, etc. Therefore, better understanding and monitoring of the occurrence of lightning in Vietnam is an important topic to examine in high detail in supporting overall lightning safety.

In Vietnam, lightning research has been pursued in PROMOSERV projects since 2010. The activities have consisted of basic introductions to thunderstorms and lightning in general, lightning location data applicability in operational weather services, and also the usability of historical data for analyzing the detailed features of the spatio-temporal occurrence of lightning in the region. Based on the experiences gained from the projects, VNMHA is now establishing a national lightning location network. In the future, when the national LLS has operated for a few years, this study can be continued by utilizing even more accurate lightning information. In addition, the research could focus on verifying the skill of forecasting the thunderstorms in the region based on, e.g., [36]. Another future research topic could be investigating the polarity, multiplicity and other lightning-specific characteristics of lightning in Vietnam.

In this study, lightning location data (a total of 315,522,761 lightning strokes) and human-observed thunderstorm days from the period 2015–2019 were analyzed to indicate the thunderstorm occurrence characteristics in Vietnam. Our results indicate that the lightning in Vietnam is most frequent during the summer monsoon and least active in the winter monsoon season. In the northern parts, a well-defined thunderstorm season is roughly from April to early October, whereas in the southern parts, the occurrence distribution has two peaks, one in the onset phase of the summer monsoon (late April–May) and one in the transition time between the summer and winter monsoon (i.e., September–October). Annually, on average, 6.9 million flashes occur in the land areas of Vietnam. However, the year-to-year variation is very large: in our study period, from about 4.6 million (in 2015) to 9.3 million flashes (in 2019). The largest amount of lightning occurs on average in May, with about 1.7 million flashes, while very little lightning is observed from December to February. On a single day, more than 200,000 flashes can be expected. On average, about 60 to 80 thunderstorm days occur annually in Vietnam, depending on the region. In the southern parts, more thunderstorm days occur in the autumn compared to the northern regions. The seasonal cycle of thunderstorm days shows similar features to those of light-

ning occurrence, i.e., peaking during the summer monsoon. The regional differences in the occurrence and intensity of thunderstorms seem to reflect the climatological characteristics of the regions: humid tropical monsoon climate in North Vietnam, moderately tropical climate with dry and rainy seasons in South Vietnam and a mix of these in the central parts of Vietnam. Our results complement historical records of [8] but provide more details of the thunderstorm day characteristics of Vietnam. However, it must be noted that a longer observational dataset should be investigated in order to provide more robust conclusions regarding the climatological characteristics of thunderstorms in Vietnam.

Finally, it can be concluded that investments in weather services and early warnings have a substantial return ratio. For example, in Finland, the ratio is 1:5 [37]. This means that the socio-economic benefits for each invested euro in the weather services produce a benefit of at least 5 euros. The ratio can even be much larger depending on what sectors are examined [37,38]. Because in practice, in all developed countries, a well-performing NHMS is an essential part of the overall national safety, the capacity building of NHMSs in developing countries should be of very high importance in support of the overall socio-economic development of the country in question (e.g., [39]).

**Author Contributions:** Conceptualization, all authors; methodology, K.V.M., T.K.L., L.P.H. and T.D.D.; software, T.K.L.; validation, A.M.; formal analysis, K.V.M., T.K.L., L.P.H. and T.D.D.; investigation, K.V.M., T.K.L., L.P.H. and T.D.D.; resources, S.K.; data curation, A.M.; writing—original draft preparation, all authors; writing—review and editing, A.M.; visualization, K.V.M. and T.K.L.; supervision, S.K.; project administration, S.K.; funding acquisition, S.K. All authors have read and agreed to the published version of the manuscript.

**Funding:** This research was funded by the Ministry for Foreign Affairs of Finland through the PROMOSERV3 project, intervention code 76909174. Lam Hoang Phuc was sponsored by the Ministry of Natural Resources and Environment (intervention code TNMT.2022.06.06). Tien Du Duc was sponsored by the Ministry of Natural Resources and Environment (intervention code TNMT.2022.06.02).

**Data Availability Statement:** Restrictions apply to the availability of these data. GLD360 data was obtained from Vaisala Inc. and are available from the authors with the permission of Vaisala Inc. The thunderstorm day observation data of Vietnam are available from the authors for research purposes.

**Acknowledgments:** We thank Vaisala Inc. for the GLD360 lightning location data. We also thank Dang Dinh Quan for his preparations of TANLs 1 and 7.

**Conflicts of Interest:** The authors declare no conflict of interest.

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
