# Peer review of "Thunderstorm Activity and Extremes in Vietnam for the Period 2015–2019"

_climate, doi:10.3390/cli10100141_

Round 1
Reviewer 1 Report (Previous Reviewer 2)
The authors have made appropriate revisions and responses to my comments. I recommend acceptance of this paper.
Author Response
Thank you for the comment and accepting our revisions. In your report, you had marked that “Extensive editing of English language and style required”. We have made a language check for our manuscript through a language translation agency. In addition, we asked a colleague to read the manuscript and check that the text is understandable, and we have made some language style changes based on the feedback.
Reviewer 2 Report (New Reviewer)
Review for article “Thunderstorm activity and extremes in Vietnam for the period 2 2015-2019”
The manuscript is generally moderate and the topic seems to present few interesting results for readers. Overall, I suggest a "major" revision before possible consideration of the application in Climate. My comments are listed as:
1. first 2 paragraphs in the introduction should be moved to another place.
2. remove "thunderstorms" from the keywords.
3. All figures are presented with a poor resolution and should be revised.
4. A language check by a professional native speaker or an editing agency is needed to fix many syntax, style and phrasing problems.
5. The manuscript need more newly references in the introduction part.
6. References style is not correct. Please revise it.
7. Please check the references as there are some references without Doi or wrongly inserted and some the location of publication is not written. Please check them carefully.
Author Response
Point 1: first 2 paragraphs in the introduction should be moved to another place.
Response 1: We have now re-organized the introduction and the first two paragraphs are moved later in the Introduction. In addition, we now include more references and have motivated our study and its novelty hopefully more clearly.
Point 2: remove "thunderstorms" from the keywords.
Response 2: We do not understand this request as the topic of the manuscript is indeed related to thunderstorms and it is additionally the first word in our title. Therefore, we have left this word to the keywords.
Point 3: All figures are presented with a poor resolution and should be revised.
Response 3: Thank you for this comment and apologies that we had not noticed this before. We agree the resolution of some of the figures was relatively poor. In the guidelines of the journal, the figures are instructed to be at least 300 dpi in resolution. This criterion was met with our original Figures 1 and 10. We have now revised all the other figures with at least 300 dpi resolution.
Point 4: A language check by a professional native speaker or an editing agency is needed to fix many syntax, style and phrasing problems.
Response 4: We have made a language check for our manuscript through a language translation agency. In addition, we asked a colleague to read the manuscript and check that the text is understandable, and we have made some language style changes based on the feedback.
Point 5: The manuscript need more newly references in the introduction part.
Response 5: We have added more background and motivation of our study to the Introduction and we added several new and recent references.
Point 6: References style is not correct. Please revise it.
Response 6: In the journal guidelines, it is stated that: “Your references may be in any style, provided that you use the consistent formatting throughout. It is essential to include author(s) name(s), journal or book title, article or chapter title (where required), year of publication, volume and issue (where appropriate) and pagination. DOI numbers (Digital Object Identifier) are not mandatory but highly encouraged.”
In these guidelines, the journal also gives an example how the references could be written:
- Author 1, A.B.; Author 2, C.D. Title of the article. Abbreviated Journal Name Year, Volume, page range.
We have followed this suggested reference style and made sure it is the same and consistent between all the citations.
Point 7: Please check the references as there are some references without Doi or wrongly inserted and some the location of publication is not written. Please check them carefully.
Response 7: We have now added DOI numbers to those citations that were missing them. In addition, we have double-checked that the publication locations are written in those citations that should include one.
Reviewer 3 Report (New Reviewer)
The article describes the climatology of lightning and flashes in Vietnam using lightning location data. It deserves to be published, although it would have been better if analysis were extended to diurnal variation of flashes. It will be acceptable after some revision.
[Main comments]
It is desired to give more description of the temporal homogeneity of the lightning location data. The authors say "all years in our dataset (2015-2019) can be considered homogeneous" (Line 102), but the logic leading to this assertion is not clearly written. The number of flashes show an increase during the five years (Fig.2), whereas thunder days do now show an increasing trend (Fig.6). It is therefore recommended to add some discussion on the possibility of temporal change of the catch efficiency of the lightning location system.
[Other comments]
Line 40 "picking up considerable moisture; consequently, the winter season in most parts of the country is dry" --- The logic of this sentence is not clear. If there is moisture supply, then the weather will be wet. It will be better to change expression.
Line 126 "A valid assumption is that the overall DE of GLD360 in Vietnam is at least 70% for cloud-to-ground flashes." --- If possible, it is better to give further evidence to support this assumption.
Line 137 "because it is sufficiently small and large at the same time" --- Please explain in what meaning the scale of 100km^2 is small, and in what meaning it is large.
Line 166 "different climatological regions (North, South, whole Vietnam)" --- It will be better to write in the text that the northern and southern Vietnam were divided by the 16N line. Also, it will be better to write the area (=size) of each region, because comparison of the number of flashes in the two areas is made in the subsequent part (Fig.3).
Line 190 "The thunderstorm season in late spring and early summer" --- Please define the seasons (spring, summer) in advance. Otherwise it will be better to describe in months.
Line 261 "It is obvious that the number and distribution of observation stations has an effect on the thunderstorm day results based on observations. Therefore, the lower number of thunderstorm days in the observations compared to the GLD360 data is expected." --- The logic of this part is not clear. Please explain in what way the low thunderstorm days in the observations are related to the distribution of stations.
Author Response
Point 1: It is desired to give more description of the temporal homogeneity of the lightning location data. The authors say "all years in our dataset (2015-2019) can be considered homogeneous" (Line 102), but the logic leading to this assertion is not clearly written. The number of flashes show an increase during the five years (Fig.2), whereas thunder days do now show an increasing trend (Fig.6). It is therefore recommended to add some discussion on the possibility of temporal change of the catch efficiency of the lightning location system.
Response 1: Before that statement, we write in the manuscript that 2,7 % of the strokes in 2019 data are classified as IC strokes. This statement about homogeneous dataset comes from our assumption that the other years include approximately the same percentage of IC strokes although the classification is not available. To hopefully clarify this logic, we have now added the following sentence before the original statement: “Although the classification is available only for 2019 data, the other years presumably include similar percentage of IC strokes.”
Point 2: Line 40 "picking up considerable moisture; consequently, the winter season in most parts of the country is dry" --- The logic of this sentence is not clear. If there is moisture supply, then the weather will be wet. It will be better to change expression.
Response 2: We agree the original sentence was not clear. We have now replaced the original sentence with “bringing cool and dry air from northeast”.
Point 3: Line 126 "A valid assumption is that the overall DE of GLD360 in Vietnam is at least 70% for cloud-to-ground flashes." --- If possible, it is better to give further evidence to support this assumption.
Response 3: The 70 % DE is stated by the GLD360 operator. This was mentioned in the same chapter earlier, but we added this information also to this sentence. We have also re-formulated this part trying to estimate the upper and lower limit of the DE based on the peak current distribution. However, because we do not have any verification dataset available, it is impossible to determine the exact DE in the study region.
Point 4: Line 137 "because it is sufficiently small and large at the same time" --- Please explain in what meaning the scale of 100km^2 is small, and in what meaning it is large.
Response 4: If the scale would be much smaller, the resulting map would be too “noisy”. On the other hand, if the scale would be much larger, it would smooth out too much details. However, we do note in the manuscript that the choice of 100 km2 is arbitrary. We have added these explanations also to the manuscript so that this statement is clearer.
Point 5: Line 166 "different climatological regions (North, South, whole Vietnam)" --- It will be better to write in the text that the northern and southern Vietnam were divided by the 16N line. Also, it will be better to write the area (=size) of each region, because comparison of the number of flashes in the two areas is made in the subsequent part (Fig.3).
Response 5: Thank you for the suggestion to add the areas of the regions. This sentence is now rewritten to explain the investigated regions more clearly. We calculated the areas for Vietnam and separately for North and South Vietnam based on the shapefile we have used for masking the lightning data in our study. These areas are now also added to the manuscript. Furthermore, as we had originally taken the surface area value (in km2) from Google Maps to calculate the annual flash density, we have now used the surface area of the shapefile. Therefore, we corrected the average annual flash count from 21 flashes / km2 / year to 20 flashes / km2 / year based on the actual area and corresponding number of strokes in our dataset.
Point 6: Line 190 "The thunderstorm season in late spring and early summer" --- Please define the seasons (spring, summer) in advance. Otherwise it will be better to describe in months.
Response 6: We have now replaced the seasons with months which we agree is more precise.
Point 7: Line 261 "It is obvious that the number and distribution of observation stations has an effect on the thunderstorm day results based on observations. Therefore, the lower number of thunderstorm days in the observations compared to the GLD360 data is expected." --- The logic of this part is not clear. Please explain in what way the low thunderstorm days in the observations are related to the distribution of stations.
Response 7: The observations are based on human observers at weather observation stations. Because the efficient observation radius for thunder by humans is around 10 km, and the distance between the observation stations is much larger than 10 km, it is very likely that these observations are an underestimation. Also, human observations may also include some random noise, e.g., one observer is more “efficient” in observing than another, etc.
We have specified the human-observable radius to Section 2.1 and additionally explain this difference between observations and GLD360 data results in more detail in lines 296-302.
Round 2
Reviewer 2 Report (New Reviewer)
Good to be published
This manuscript is a resubmission of an earlier submission. The following is a list of the peer review reports and author responses from that submission.
Round 1
Reviewer 1 Report
Thunderstorm characteristics and extremes in Vietnam for the period 2015-2019
The paper discussed the variation of lightning activity in Vietnam using the global lighting location system maintained by the Visala. The information provided by the authors is helpful for the scientific community. However, the information provided in the paper should not be accepted without a major revision. My observations and comments regarding the paper are given below.
First, the title of the paper does not match the content. The paper discussed the lightning activity variation from 2015 to 2019. It does not discuss thunderstorm characteristics.
From lines 62 to 88, the authors discussed the project that supports this study. It does not provide any scientifically important information.
The authors claim that the lightning density in Vietnam is 21 flashes/kilometre/year. However, the Visala home page provides a much higher value of lightning flash density in Vietnam. Both publications use the same data provided by Visala global lightning location system. The authors have not provided any clarification for the discrepancy.
According to figure two of the paper, between the years 2015 to 2019, the number of lightning flashes that strike in Vietnam increased considerably. However, as shown in figure 6 no increase in thunderstorm days per year was observed. The authors have not discussed The reasons for the increasing number of lightning per thunderstorm.
From lines 305 to 308, the authors claim that the detection efficiency of the lightning locking system is lower. But as shown in Figure 10, the lightning location system has detected lightning with lower current amplitude. This observation does not support the claim of the authors.
Reviewer 2 Report
The authors analyzed lightning activity and thunderstorm characteristics over Vietnam from 2015 to 2019. This paper reveals the characteristics of lightning activity over southern and northern Vietnam in recent years, which has certain reference value. Here are some comments for the authors to address.
1. Whether the lightning location data used in this paper is the total lightning detected or cloud-to-ground lightning, and whether quality control is carried out for positive lightning and negative lightning. Especially for the positive lightning, if there is no quality control, the conclusion of the paper is questionable.
2. Currently, there are not many studies on lightning in Vietnam, but there are many studies on lightning in India and China. It is necessary to supplement the previous studies on lightning activities in nearby areas in the introduction, which is conducive to further discussion on lightning activities in Vietnam. It is suggested to cite the following papers (not limited to):
Dewan, A., Ongee, E.T., Rahman, M.M. and Mahmood, R. (2018) Lightning activity associated with precipitation and CAPE over Bangladesh. International Journal of Climatology, 38, 1649–
1660. https://doi.org/10.1002/joc.5286.
Iwasaki, H. (2016) Relating lightning features and topography over the Tibetan plateau using the world-wide lightning location network data. Journal of the Meteorological Society of Japan Ser II, 94(5), 431–442.
Xia, R., Zhang, D. and Wang, B. (2015) A 6-yr cloud-to-ground lightning climatology and its relationship to rainfall over central and eastern China. Journal of Applied Meteorology and Climatology, 54(12), 2443–2460. https://doi.org/10.1175/JAMC-D-15-0029.1.
Zhao, P., Xiao, H., Liu, C., Zhou, Y., Xu, X., & Hao, K. (2022). Evaluating a simple proxy for climatic cloud-to-ground lightning in Sichuan Province with complex terrain, Southwest China. International Journal of Climatology, 42(7):3909–3927. https://doi.org/10.1002/joc.7451
3. What is the climatic background of Vietnam, especially the meteorological elements that influence lightning activity, such as thermodynamic conditions, etc. Especially in the southern Vietnam and northern Vietnam, how different their climate background is.
4. Figure 10 shows the frequency of positive and negative lightning peak currents, which is not enough to reflect the activity characteristics of positive and negative lightning in Vietnam. Please add the spatial and temporal distribution characteristics of positive and negative lightning density in Vietnam.
5. Before discussing positive and negative lightning, authors should discuss why positive lightning is discussed. It is because positive lightning is more dangerous. Some factors affecting lightning polarity should also be added. It is suggested to cite the following papers (not limited to):
A. Nag and V. A. Rakov, Positive lightning: an overview, new observations, and inferences, Journal of Geophysical Research: Atmospheres, vol. 117, D8109, 2012.
E. R. Mansell, D. R. MacGorman, C. L. Ziegler, and J. M. Straka, Charge structure and lightning sensitivity in a simulated multicell thunderstorm, Journal of Geophysical Research, vol. 110, no. D12, D12101, 2005.
Zhao, P., Xiao, H., Liu, C., Zhou, Y. (2021) Dependence of Warm Season Cloud-to-Ground Lightning Polarity on Environmental Conditions over Sichuan, Southwest China. Advances in Meteorology, 1500470, https://doi.org/10.1155/2021/1500470
Reviewer 3 Report
This work presents the spatial and temporal distribution of lightning in Vietnam based on the GLD360 network. To the best of my knowledge, few relevant references have reported this and the topic is interesting to the lightning community. This paper is well written, with clear figures to support the paper’s conclusions. I would like to recommend this work publish after considering some suggestions as follows.
1., the authors should give a discussion of why thunderstorms in South Vietnam are generally more intense.
2. The GLD data could also give the identification of CGs and IC. Can you show the distribution of CGs and IC, respectively.